# Patterns of sexual violence against adults and children during the COVID-19 pandemic in Kenya: a prospective cross-sectional study

Sarah Rockowitz [iD],[1] Laura M Stevens,[1] James C Rockey,[2] Lisa L Smith,[3] Jessica Ritchie,[3] Melissa F Colloff,[1] Wangu Kanja,[4] Jessica Cotton,[1] Dorothy Njoroge,[5] Catherine Kamau,[4] Heather D Flowe [iD] [1]

[1]School of Psychology, University of Birmingham, Birmingham, UK
[2]Birmingham Business School, University of Birmingham, Birmingham, UK
[3]School of Criminology, University of Leicester, Leicester, UK
[4]Wangu Kanja Foundation, Nairobi, Kenya
[5]Department of Journalism & Corporate Communication, United States International University, Nairobi, Kenya

**Correspondence to**
Dr Heather D Flowe;
h.flowe@bham.ac.uk

## ABSTRACT

**Objectives** This study examined patterns of sexual violence against adults and children in Kenya during the COVID-19 pandemic to inform sexual violence prevention, protection, and response efforts.

**Design** A prospective cross-sectional research design was used with data collected from March to August 2020.

**Setting** Kenya.

**Participants** 317 adults, 224 children.

**Main measures** Perpetrator and survivor demographic data, characteristics of the assault.

**Results** Bivariate analyses found that children were more likely than adults to be attacked during daytime (59% vs 44%, p<0.001) by a single perpetrator rather than multiple perpetrators (31% vs 13%, p<0.001) in a private as opposed to a public location (66% vs 45%, p<0.001) and by someone known to the child (76% vs 58%, p<0.001). Children were violated most often by neighbours (29%) and family members (20%), whereas adults were equally likely to be attacked by strangers (41%) and persons known to them (59%). These variables were entered as predictors into a logistic regression model that significantly predicted the age group of the survivor, $\chi^2$(5, n=541)=53.3, p<0.001.

**Conclusions** Patterns of sexual violence against adult and child survivors during the COVID-19 pandemic are different, suggesting age-related measures are needed in national emergency plans to adequately address sexual violence during the pandemic and for future humanitarian crises.

## STRENGTHS AND LIMITATIONS OF THIS STUDY

⇒ This study was conducted in partnership with front-line, human rights defenders, survivor-led organisations/networks and social justice centres using a prospective study design, which enabled the systematic and rapid study of sexual violence in Kenya during the pandemic, even though there were considerable physical distancing measures in place.

⇒ The data provide detailed information about survivors and perpetrators, including where and when incidents occurred, which enabled us to compare patterns of sexual violence in adults and children.

⇒ The sample comprised individuals who were seeking help in accessing vital services; therefore, inferences about patterns nationally in Kenya cannot be made because the data may not be representative.

⇒ Information about whether patterns of sexual violence are changing during the pandemic remains unknown because sexual violence is under-reported and also more difficult to report and document during emergencies/humanitarian situations/pandemics, and there is a need for real-time data collection systems that gather and analyse detailed, longitudinal information about sexual violence especially in resource-limited countries like Kenya, where service and response infrastructure are not as robust.

## INTRODUCTION

This study focuses on Kenya, a country that has a long history of sexual and gender-based violence (SGBV), which is exacerbated during times of national crisis, such as during election periods.[1] While SGBV affects women, men and children, it disproportionately affects women and girls, with one in three women having faced SGBV in their lifetimes worldwide.[2] Previous conflicts and disasters have led to increased gender inequality, gender-based violence and other human rights violations, owing to disruptions in response (medical), protection and legal services.[3] The arrival of COVID-19 in Kenya in early March 2020 marked the start of another national crisis, with more than 56 000 confirmed cases as of November 2020.[3–5] In late March, President Kenyatta issued a nationwide curfew, with all non-essential travel banned between 19:00 and 05:00. Schools and non-essential businesses had to close, and travel in and out of the country was heavily restricted.[4 5] These measures have been extended and modified multiple times, and by November 2020, the number of people who could gather in

groups was still limited. Schools were also still closed, but universities opened, and air travel restrictions were lifted.[6] While the measures have undoubtedly curbed the spread of the disease, they seemed to be compromising the safety and well-being of citizens. In particular, there have been widespread reports of increases in domestic and sexual violence during the COVID-19 crisis.[5 6]

Around the world, humanitarian crises, such as natural disasters, conflict and disease outbreaks, are associated with changing patterns of sexual violence.[7–9] After the 2010 earthquake in Haiti, for example, the odds of an adolescent girl in Haiti being sexually abused increased by 41%.[9] Increased sexual violence occurs during conflicts, notably in Rwanda, Kosovo and the Democratic Republic of the Congo (DRC). These crimes are especially prevalent against women and children, and attacks by multiple perpetrators are common. In the DRC, for example, nearly 76% of women surveyed had experiences of rape that were consistent with the attack being used as a weapon of war, and 69% of women reported experiencing gang rape, with these incidents typically being perpetuated by three perpetrators on average.[10] These findings are consistent with research conducted in the Central African Republic, Libya and Mali, which found that multiple perpetrator rape was commonly reported by survivors.[11]

SGBV increases during disease outbreaks, with studies reporting increases in Sierra Leone, Liberia and Guinea during the Ebola outbreaks in West Africa in 2014–2016, and especially high increases in teenage pregnancies were reported in Sierra Leone.[12 13] Similarly, Zika and cholera outbreaks have been linked with increased incidence of domestic violence, and reductions in funding for and access to public health services.[14] Physical distancing measures implemented during pandemics are also thought to be responsible for changing patterns and increases in violence. For instance, lockdowns and curfews mean that people must remain indoors with abusers and are unable to access outside assistance because police and vital services are unavailable, and abusers can act with impunity.[15 16]

More research on SGBV during times of compounding crises is needed, however. SGBV is highly stigmatised, which leads to under-reporting, especially in resource-limited countries that have high levels of gender inequality. Further, it is difficult to assess whether patterns and rates of SGBV are changing during times of crisis, owing to the unavailability of nationally representative data and a lack of up-to-date and recurring data collection, as well as a lack of data harmonisation, which would allow for examining SGBV trends in relation to humanitarian crises, and inform effective prevention, protection, and responses.

During the ongoing COVID-19 outbreak, several months of lockdown measures, economic challenges, health concerns and changing global relations have increased concerns of a heightened risk of SGBV. This violence during lockdown is being considered a shadow pandemic with the United Nations Population Fund

estimating an additional 31 million cases of SGBV worldwide following 6 months of isolation.[16 17] Governments in some countries have had to create or supply alternate housing for people fleeing abusive situations, as was the case in Italy and France, with hotels being used as safe houses.[16] Social isolation policies have distinct impacts on children as well. Adolescent girls' absence from school, coupled with the lack of alternative safe spaces or shelters, has been associated with increased vulnerability to sexual violence from family members and others, including guardians, neighbours, and other community members.[18] As seen during the Ebola crisis, the closure of schools was associated with increased sexual violence against girls and boys, child pregnancies, and child marriage.[19]

This study prospectively investigated patterns of sexual violence perpetrated against adults and children in Kenya during the COVID-19 pandemic. We analysed data from interviews with adult survivors and the guardians of child survivors conducted by human rights defenders and members of the social justice centres during the pandemic. We focus on sexual violence because it has received less attention during the pandemic compared with physical violence. Further, research to date has not compared patterns of violence for adults and children. Doing so is critical because social isolation measures may differentially affect people in relation to age, and different measures may need to be put in place depending on the age group to prevent and respond to SGBV during COVID-19.

Based on the literature reviewed above, we predicted that there would be a greater number of women and girls violated compared with men and boys. Additionally, we anticipated there would be age-related differences in the types of locations in which sexual violence is occurring. Owing to school closures, and a lack of alternative safe spaces, we predicted that children would be at a greater risk than adults during the day, and in private compared with public locations. We also compared the incidence of multiple versus single perpetrator attacks to better understand the nature of the violence occurring in relation to age. To our knowledge, no previous research has compared adults and children regarding the prevalence of violations committed by multiple perpetrators in Kenya. Hence, no age group predictions were made concerning multiple perpetrators.

## METHOD
### Design
A quantitative between-group prospective research design was used. The criterion variable was age group (child or adult survivor). The predictor variables included the offence characteristics displayed in table 1, which also summarises how the variables were operationalised.

### Participants
Participants (n=787) were survivors of sexual violence. All were residents of Kenya, living in 23 counties and

**Table 1** Descriptions of how predictor variables were coded and operationalised

| Variable | Definition |
|---|---|
| Female survivor | Whether the survivor was female (coded as 1) as opposed to male (coded as 0). |
| Male perpetrator | Whether the perpetrator was male (coded as 1) as opposed to female (coded as 0). |
| Daytime attack | Whether the attack occurred in daytime (06:00–17:59; coded as 1) as opposed to at night (18:00–05:59; coded as 0). |
| Private or public location | Whether the attack occurred in a private home (coded as 1) as opposed to a public location where the violation could have been witnessed or interrupted by a member of the public (coded as 0). |
| Private location type | Private locations were further subdivided into victim residence (coded as 1 for victim residence, 0 for any other location public or private); perpetrator residence (coded as 1 for perpetrator residence, 0 for any other location public or private); or other residence (coded as 1 for other residence, 0 for any other location public or private). |
| Multiple perpetrator | Whether the attack was perpetrated by more than one perpetrator (coded as 1) as opposed to a singular perpetrator (coded as 0). |
| Known or stranger perpetrator | Whether the attack was perpetrated by someone known to the survivor (coded as 1) or a stranger (coded as 0). |
| Perpetrator relationship type | Perpetrator relationship type was subdivided into neighbour (coded as 1 for neighbour, 0 for any other relationship type); stranger (coded as 1 for stranger, 0 for any other relationship type); family member (coded as 1 for family member, 0 for any other relationship type); acquaintance/friend (coded as 1 for acquaintance/friend, 0 for any other relationship type); spouse/husband/boyfriend (coded as 1 for spouse/husband/boyfriend, 0 for any other relationship type); authority figure (coded as 1 for authority figure, 0 for any other relationship type); or other (coded as 1 for other, 0 for any other relationship type). |

had contacted human rights defenders for assistance in obtaining vital services in the aftermath of sexual violence during the COVID-19 pandemic between March and August 2020. The human rights defenders interviewed the survivors (or their legal guardians if they were under 18) about the offence on intake. The interview protocol was informed by the WHO's ethical principles for research on SGBV and safety protocols developed by the human rights defenders for conducting their work with survivors. The survivors were aged between 7 months and 72 years (M=21.3; SD=9.4 years).

Survivors were categorised into two age groups. Following definitions provided by the WHO, the child group included survivors aged 17 years and younger, whereas the adult group included survivors aged 18 years and older.[1]

After excluding cases with missing data on the predictor variables, the final sample consisted of 224 survivors in the child group and 317 in the adult group. The participants in the final sample for the child group ranged in age from 8 months to 17 years (M=12.6, SD=3.9), 83% were girls and 93% were perpetrated against by men, and for the adult group, participants ranged in age from 18 to 72 (M=27.1, SD=8.1) years, 92% were women and 96% were perpetrated against by men.

### Materials
The data were obtained from records held by the human rights defenders who were assisting survivors in accessing vital services during the pandemic. They interviewed survivors about the incident and recorded information about the case on their standard intake form (online

supplemental file 1). They recorded the date, time and location of the incident, and gave a free text description summarising the incident. The form also had specific items to document the number of perpetrators, the relationship between the survivor and perpetrator(s), the location of the attack and the age and gender of the survivor and perpetrator. Additionally, while not analysed in the current paper, any services (eg, police, medical, safe house) the survivor had accessed were also recorded.

### Procedure
Each intake form was read by two members of the research team to create the data set. They coded the data using the criteria outlined in table 1. If there were missing data on the form, the team read the incident summary and attempted to complete the missing information.

### Ethics
The confidentiality of the data was maintained by the research team, and safety precautions were taken to minimise any risks that might cause physical harm to participants of this study. Data collection involved qualitative interviews only, and participants were offered psychological services after interview. The Kenyan Data Protection Act (2019) was adhered to in the conduct of this research study.[20] Special attention was paid to Part IV of the Act, which notes that personal data should be processed with special attention to the privacy of the data subject, data should only be collected for specified and legitimate purposes and that the data subject has a right to know how their data are used.[20] The data belong to the Wangu Kanja Foundation and the Sexual Violence

Survivors' Network in Kenya, and permission to use the data was obtained from these organisations to conduct the analyses.

## Patient and public involvement

We relied heavily on input from civil society grass-roots organisations who work on the frontlines to assist survivors in accessing vital services in the aftermath of sexual violence, including the Wangu Kanja Foundation and the Sexual Violence Survivors' Network in Kenya. These organisations codeveloped the research questions, the study design including the data collection instruments. They also conducted participant recruitment, data collection and assisted with manuscript preparation. Their experience and knowledge with sexual violence in Kenya informed every aspect of the project. Their reputation within the Kenyan communities enabled survivors to disclose the incidents that occurred. The Wangu Kanja Foundation and human rights defenders would also be integral in disseminating the research findings to their networks and relevant stakeholders.

## Statistical analysis

As our main analysis, we used logistic regression with age group as the dependent variable to determine which offence characteristics significantly differentiated the child and adult groups. The child age group was coded as 1 in the analysis, whereas the adult age group was coded as 0. While our data contain detailed information about each attack, we restrict our analysis to a limited number of binary variables in table 1 as predictors. This is because there is a risk of a statistical common support problem if we use a finer grained analysis. For example, while we have detailed data on where survivors were attacked, or their relationship with the perpetrator, we could not exploit this as few in the child group were attacked going to work, or by their spouse or partner. To avoid this difficulty, we used the coarser coding of relationship, known versus stranger perpetrator and whether the attack took place in a public place or in private. This ensured that there were sufficient numbers of both children and adults in all categories. We then supplemented this analysis by tabulating the finer coding in tables 2 and 3.

**Table 2** Distribution of perpetrator relationship to survivor within age group

| | Child n=224 (%) | Adult n=317 (%) |
|---|---|---|
| Neighbour | 29 | 6 |
| Stranger | 25 | 41 |
| Family member | 20 | 5 |
| Other | 12 | 16 |
| Acquaintance/friend | 11 | 12 |
| Spouse/husband/boyfriend | 3 | 15 |
| Authority figure | 2 | 6 |

**Table 3** Distribution of attack location within age group

| | Child n=224 (%) | Adult n=317 (%) |
|---|---|---|
| Perpetrator's house | 41 | 20 |
| Public | 28 | 48 |
| Survivor's house | 23 | 23 |
| Other house | 7 | 6 |
| Survivor/perpetrator's house | 1 | 3 |

We conducted preliminary analyses to identify which variables to enter into the model using Pearson's $\chi^2$ tests for association. This allowed for testing whether the association between age group and each of the dichotomous variables was statistically significant. Only the variables that were significantly associated with age group were entered into the logistic regression model. To control for type 1 errors, Bonferroni corrections were applied to the 0.05 alpha level (adjusted alpha=0.008, with six variables). The strength of the relationship between the individual offence characteristics was assessed using Cramer's V, which measures the magnitude of the relationship between two categorical variables.[21] Values that fall between 0.41 and 0.60 were interpreted as large, whereas values that fall between 0.20 and 0.40 were interpreted as moderate in magnitude. Values smaller than 0.20 were regarded as associations small in magnitude. All analyses were conducted using SPSS V.26.

Our data are freely available at: https://osf.io/b9dzp/.

## RESULTS

Bivariate ($\chi^2$) analysis indicates that children compared with adults were less likely to be female and less likely to be attacked by multiple perpetrators (table 4). Regardless, both child and adult victims were overwhelmingly female; 83% and 92%, respectively. Children were also more likely to be attacked in a private location by a known perpetrator and were more likely to be attacked in the daytime. The associations between age and private location, and age and multiple perpetrators were moderately large, whereas the strength of the other associations, while statistically significant, was small in magnitude.

In the logistic regression model, the variables that were statistically significant from the bivariate analysis ($\chi^2$ results) were entered as predictors (ie, female survivor, daytime attack, private vs public location, multiple perpetrators and known vs stranger perpetrator), and the dependent variable was age group. The results are shown in table 5. The overall model was statistically significant, $\chi^2$(5, n=541)=53.3, p<0.001. According to Nagelkerke's $R^2$, 13% of the variability in age group was accounted for by the predictors in the model. Child compared with adult survivors were 1.61 times more likely to be attacked during the day, and 1.72 times more likely to be attacked in private as opposed to in public. Child compared

**Table 4** Comparisons between characteristics of sexual violence against children versus adults: bivariate analysis

| Variable | Child n=224 (%) | Adult n=317 (%) | Pearson's $\chi^2$ | P value | Cramer's V |
|---|---|---|---|---|---|
| Female victim | 83 | 92 | 11.41 | 0.001 | 0.145 |
| Male perpetrator | 92 | 94 | 1.17 | 0.279 | 0.047 |
| Daytime attack | 59 | 44 | 13.18 | <0.001 | 0.156 |
| Private versus public location | 66 | 45 | 21.55 | <0.001 | 0.2 |
| Multiple perpetrators | 13 | 31 | 24.2 | <0.001 | 0.212 |
| Known versus stranger perpetrator | 76 | 58 | 17.86 | <0.001 | 0.182 |

with adult survivors were also significantly less likely (OR=0.458) to be female, and less likely (OR=0.528) to be attacked by multiple perpetrators.

Tables 2 and 3 present a more detailed descriptive analysis of the child and adult cases on the relationship between the perpetrator and the victim, and the locations in which the attacks took place. As can be seen, age group was significantly associated with the relationship between the perpetrator and the survivor, $\chi^2(7, n=541)=107.84$, p<0.001. Children were most often victimised by neighbours, followed by strangers, and family members, whereas adults were most often victimised by strangers, followed by other types of perpetrators (customer, community member, friend of a friend), and spouses. Age group was also significantly associated with attack location, $\chi^2(4, n=541)=35.59$, p<0.001. Children were most often attacked at the perpetrator's house (41% of the cases), whereas adults were most often attacked in public locations (48% of cases).

## DISCUSSION
### Summary of key findings
We compared patterns of sexual violence committed against adults and children in Kenya during the COVID-19 pandemic. The data arose from interviews conducted by human rights defenders with survivors and describe the experiences of 541 survivors. We found that the children in our sample were on average 4 years younger compared with national surveys of children in Kenya.[22] [23] Further,

**Table 5** Outputs of logistic regression by predictor variable

| | df | Estimate | SE | Wald $\chi^2$ | P value |
|---|---|---|---|---|---|
| Female victim | 1 | −0.782 | 0.29 | 7.3 | 0.007 |
| Daytime attack | 1 | 0.474 | 0.19 | 6.5 | 0.011 |
| Private versus public location | 1 | 0.543 | 0.21 | 6.96 | 0.008 |
| Multiple perpetrators | 1 | −0.638 | 0.27 | 5.77 | 0.016 |
| Known versus stranger perpetrator | 1 | 0.295 | 0.34 | 1.6 | 0.21 |

compared with adults, children were more likely to be attacked during the day, in private as opposed to public locations, by lone perpetrators and by neighbours. In what follows, we discuss our findings in relation to existing research and draw implications for policy.

### Comparisons to current literature
There were significant numbers of children in our sample, which is unsurprising, as approximately half of gender-based violence (GBV) survivors are children during humanitarian crises. However, the children in our sample were 12 years old on average, which is 4 years younger than the nationally representative samples taken before the pandemic.[22] [23] Our sample was not nationally representative due to time and resource constraints, and it must also be noted that this violence was occurring within a particular crisis and the consequences of other crises may be different. However, it is still notable that the child survivors in our sample were younger than previous national samples have indicated. A recent study in Kenya noted that survivors who are being seen in medical settings during the pandemic appear to be younger compared with before the pandemic, speculating this is due to school closures during the pandemic.[24] Although SGBV, such as domestic violence, has been linked to cases of domestic homicide in Kenya, there were no mortalities captured in our study sample.[25]

We also found that children were 1.61 times more likely than adults to be attacked during the day. This could be attributed to the way that children and adults were spending their time during the pandemic. Because schools were closed, and there was no provision of any alternative safe spaces, children may have been often left alone or under the care or supervision of neighbours or community members, which may have made them more vulnerable to attack in some instances. Children were more likely to be attacked in private as compared with public locations. Adults in our sample were about equally likely to be attacked during the day as at night. Further, in keeping with previous research, significantly more adults were violated by multiple perpetrators in one attack compared with children.[26]

The proportion of boys in the child group was larger than the proportion of men in the adult group. This

may reflect differential rates of victimisation for men compared with boys, as boys are more vulnerable to assault than men due to their age. Another possible reason is that sexual violence against men compared with boys is disclosed less often. The legal definition of rape in Kenya, like many countries, requires 'vaginal penetration', which reinforces sociocultural notions that men cannot be sexually victimised.[27] Further, the tendency for people to believe that the victimisation of men is harmless, coupled with self-blame, and fear on the part of victims that their community and family will react negatively towards them, discourages men from seeking help, and reporting sexual offences to the police.[28]

The children in our sample were more likely to be violated by someone they knew than a stranger. For adults, perpetrators were most likely to be strangers, followed by neighbours and community members, and spouses. The most common perpetrators for children were neighbours, followed by strangers and family members. Adults were violated by strangers more frequently because they were often attacked when the opportunity struck, such as while walking to or from work, whereas children were violated by neighbours when they were left under their supervision due to school closures and their parents' job requirements.[29] Children were attacked by neighbours and in the perpetrators' houses at higher rates than adults. Although both groups were more likely to be violated by someone they knew as opposed to a stranger, and in both groups more than half of the perpetrators were known to the survivor, there is a high proportion of strangers compared with known assailants in both age groups. There were several instances in our data set in which neighbours invited children to use a computer or access the Internet, and then assaulted them once they were inside the neighbour's residence.

## Strengths and limitations

This research was conducted in partnership with frontline, survivor-led organisations using a prospective study design, which are key strengths. This enabled us to study sexual violence systematically and rapidly in Kenya during the pandemic, even though there were considerable physical distancing measures in place. Further, our data are unusually rich. The data provide detailed information about survivors and perpetrators, where and when the incidents occurred, which allowed for studying patterns of sexual violence. There are also several limitations of our study to note. First, the sample comprised individuals who were seeking help in accessing vital services. Hence, inferences about patterns nationally in Kenya cannot be made because the data may not be representative. Further, our data do not provide information about overall sexual violence trends in our study setting. Like many countries around the world, sexual violence is under-reported, and detailed, longitudinal information about sexual violence incidents is lacking. Consequently, researchers struggle to make inferences about whether patterns of violence are changing during COVID-19.[30] For example,

the Demographic and Health Survey in Kenya, which is a nationally representative survey of adults conducted every 5 years, does not gather in-depth information about violations. For example, it does not collect information about the time of day (or night) the attack occurred, if there were multiple perpetrators involved in the attack and if the attack took place in a public or private location. Similarly, the national Violence Against Children Survey in Kenya, conducted in 2010 and 2019, does not gather in-depth information. We also had to rely on secondhand accounts from guardians of sexual violence against children, owing to a lack of trained personnel and adequate resources in Kenya for interviewing children. Some children were too young to be interviewed, with the youngest victim being just 7 months old. Finally, our model better accounted for patterns of violence against adults compared with children. This is because some of the factors we analysed were more applicable to adults than children (eg, employment, romantic relationships, being alone in public).

## Recommendations and conclusion

We urge policymakers to ensure that government COVID-19 emergency management and recovery planning adequately addresses SGBV and that minimising the risk of additional SGBV risk is integrated into national crisis policies. In particular, the results above suggest this should include the provision of adequate alternative safe spaces and shelters when schools are closed. Further, many communities have voluntarily organised neighbourhood watch groups that are focused on security issues, and these should be explicitly expanded and supported to monitor and prevent SGBV. Community leaders have also said that there is a need for more social halls—community facilities for holding meetings, which would enable screening educational films, and other social activities. These structures can be a safe space for children and can be built using constituency development funds, which each member of parliament in Kenya receives to undertake projects that will address the urgent needs of their constituents.

Our results indicate the importance of high-quality and timely data in understanding and thus combating SGBV. We thus recommend governments invest in real-time data collection and analysis systems to capture the evolving distribution of SGBV and to allow for the study of regional trends. Data collection would allow authorities to identify crime hotspots and violations being perpetrated by serial offenders, and to monitor the accessibility of vital services to help ensure that survivors have support. This information is crucial in designing effective interventions. For example, by knowing the location and time of attacks, there can be more vigilance and awareness of SGBV against children. Additionally, this information can be used to provide further education about SGBV against children and can highlight signs to look out for of abuse. These interventions can be low cost, with communities mobilised to create such activities with the

help of university students, local non-governmental organizations (NGOs), neighbourhood teachers and religious organisations. Police patrols and community initiatives could also be planned for times at which SGBV rates peak to deter attacks and apprehend offenders. Further, the installation of street lighting might deter perpetrators from attacking women and children. Another suggestion is to establish a national sexual offender register in Kenya that would warn communities about high-risk offenders. The collection of real-time data can also inform educational programmes that sensitise parents and children about community risks. These efforts must be survivor-centred, involving survivors in the implementation and evaluation of the systems.

More generally, the results in this paper highlight the latent risk of SGBV, particularly for women and girls. While its manifestation currently waxes and wanes dependent on the context, meaningful reductions in violence will require changing the narrative such that SGBV is understood to be a crime, a gross violation of human rights, and that its pre-eminent importance as a determinant of physical, emotional and mental health is reflected in national and county budgets. Funding for programming, interventions and research should be included.

High rates of SGBV also necessitate adequate protection for the needs of survivors. To this end, the national government has approved the use of the National Government Affirmative Fund to facilitate the establishment of safe spaces/shelters in all 47 counties to ensure survivors' safety and security is safe guarded. However, advocacy is required to ensure the funds are directed appropriately.

The implementation of emergency referral pathways that enable survivors to access comprehensive care and support services should be enacted by the government. Curfews and other social distancing regulations need to include SGBV response mechanisms to ensure the continued availability and accessibility of services for survivors. Further, the medicolegal response to SGBV can be strengthened by expediting restraining orders and prosecutions, and by establishing 'one-stop' centres to allow survivors to access essential services, and authorities to collect evidence, all in one location. This would also facilitate the preservation of evidence and protection of survivors to facilitate access to justice.

**Acknowledgements** We are grateful to the many necessarily anonymous human rights defenders who continue to work with survivors in difficult and dangerous conditions in Kenya.

**Contributors** HDF and WK conceptualised and designed the data collection methods used in this project. SR and LMS drafted the protocol which was then reviewed by HDF, MFC, JCR and LMS. HDF, SR and LMS analysed the data. HDF, SR, LMS, JCR, MFC, LLS, JR, WK, JC, DN and CK contributed to and reviewed the draft version of the manuscript and approved the final version.

**Funding** This research was funded by ESRC Grant ES/T010207/1 (to HDF, MFC, JCR and the Wangu Kanja Foundation) and AHRC Grant AH/T008091/1 (to HDF and the Wangu Kanja Foundation), the Institute for Global Innovation, University of Birmingham (to HDF and the Wangu Kanja Foundation), and a research grant from the Ring for Peace Foundation (to LLS).

**Competing interests** None declared.

**Patient consent for publication** Not required.

**Ethics approval** This research was approved by the Science, Technology, Engineering and Mathematics Ethics Committee of the University of Birmingham (ERN_19-0183).

**Provenance and peer review** Not commissioned; externally peer reviewed.

**Data availability statement** Data are available in a public, open access repository. All data generated during this study will be included in the published scoping review and will also be made available upon request.

**ORCID iDs**
Sarah Rockowitz http://orcid.org/0000-0002-2759-8052
Heather D Flowe http://orcid.org/0000-0001-5343-5313

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
