## [Reviewer comments · BMJ Open]

ARTICLE DETAILS

TITLE (PROVISIONAL)	Patterns of Sexual Violence Against Adults and Children during the COVID-19 Pandemic in Kenya: A Prospective Cross-Sectional Study
AUTHORS	Rockowitz, Sarah; Stevens, Laura M.; Rockey, James; Smith, Lisa; Ritchie, Jessica; Colloff, Melissa; Kanja, Wangu; Cotton, Jessica; Njoroge, Dorothy; Kamau, Catherine; Flowe, Heather ;

VERSION 1 – REVIEW

REVIEWER	Roseboom, Tessa Amsterdam UMC Location AMC, Obstetrics and Gynaecology
REVIEW RETURNED	11-Mar-2021

GENERAL COMMENTS	This is an important paper describing a worrying trend of increasing violence due to the measures that aim to prevent the spread of the Corona virus with the unintended side effect of increasing violence against children and adults. The paper is well written and highlights the importance of more policies and protection measures to prevent any further damage to current and future generations due to the increase in violence. I fully support the authors plea but have several suggestions to improve the impact of the paper. First, the introduction highlights the gender inequality in violence exposure during crises. This is a relevant point but makes the reader think the paper will only address violence against women and children, while in actual fact the paper also addresses violence against men. Therefore I suggest to rewrite the introduction to make clear that both men and women can become victims of violence and that there is a gender equality. Also, in rewriting the introduction, please make it more concise and shorter. The authors report a clear increase in reports of violence but do not really address the extent to which this reflects both an actual increase in violence or also additionally a change in reporting. Are there any indications that there is less reporting so that the increase reported is only the tip of the iceberg of actual violence acts?
--

	The reporting of violence is only one measure of violence occurring and some victims of violence actually die. So in addition to reporting the numbers of reports about violence it would be worthwhile to also mention any mortality due to violence in this report to get a more complete picture of violence in the country. Additionally, I would urge the authors to write more about what could be done to prevent a further escalation of violence. Not only paying attention to the victims but also trying to PREVENT further violence from happening. There should be mentioning of policy interventions that could contribute to preventing violence from occurring. In the statistical methods, the authors report about the predictive ability of their models. This seems rather inappropriate because the models were not built to predict violence, but merely to explain differences in prevalence of violence according to age, gender and perpetrator, so please remove the comments about the predictive value. This is an important paper that could gain power if the above suggestions are addressed.
--	--

REVIEWER	Khurana, Bharti Harvard University, Radiology
REVIEW RETURNED	13-Mar-2021

GENERAL COMMENTS	Well done. Recommendations can be expanded, especially on alternative safe venues when schools are closed.
--

REVIEWER	Sam-Agudu, N Institute of Human Virology
REVIEW RETURNED	29-Jun-2021

GENERAL COMMENTS	This is a quite well-written paper that evaluates patterns of sexual violence against both adults and children in Kenya during the early days of the COVID-19 pandemic and response. GENERAL: 1. My major issue with this paper is the authorship credit. There is a problem at first glance, where for a study done exclusively in Kenya, one only sees one Kenyan/Kenya affiliated co-author, and in the 8th of 10 positions in the byline. Everyone else is from the UK. I would like the authors to reconsider the local contributing research staff included for authorship credit. The authors have described a cohesive consultative and implementation partnership between UK and Kenya-based staff. From the manuscript narratives, one can see that local Kenyan staff made significant contributions, at the very least in design and data collection.” Please let more of that reflect in the author byline; more of the key contributing Kenyan staff may be invited to further contribute to the requested manuscript revisions, provide further critical review, and approve the final version to be submitted.
--

The authors themselves stress on how heavily they relied on the local advocacy and sexual violence survivor service/support groups in designing the tools and implementing the study. It is not clear to what extent these local groups or individual representatives) contributed to analyzing, interpreting and reporting the data, and whether they were invited to do so. However, the level of contribution of these local groups is quite obviously high. Authors also report that the data belong to the local WK foundation, if so, how come many more members of this data-providing foundation or key local Kenyans consulted did not contribute enough to warrant authorship, but several other people from the UK did?

2. There is no mention of the study period the reported data relates to; I have looked for it, and see none. At what period in the pandemic did the reports occur and the data collected? It is critically important, and should be clearly stated in the Abstract as well as the Methods section.

TITLE: Suggest addingpatterns of **sexual" violence...The word sexual is critical

ABSTRACT

1. Design-add dates of study period
2. Results-add some actual hard data and stats, rather than simply describing. Sometimes the abstract is the only thing your audience will read.
3. "On average, the children in the sample were four years younger compared to the average age reported in national samples pre-pandemic (age 12 versus 16). Survivors were more likely to be female than male." Given that the study sample is not nationally representative, this statement has to be taken with a large grain of salt (and the authors have said so themselves)-as such, it really should not be herein the abstract. Please delete and use the space to provide harder evidence that was statistically significant and less vulnerable to the study's limitations.
4. Conclusions-... during the pandemic **and for future humanitarian crises**.

STRENGTHS AND LIMITATIONS

1. Lines 26-31-apt statement, agree, please correct "a representative" to just "representative" (delete "a"). Do the same for other instances where this shows up in the paper-about 2-3 more times
2. Line 36 to 43: This statement is true for everywhere in the world regardless of economics, may be more pressing in LMICs, so I suggest rephrasing: ..."about sexual violence **especially in countries like Kenya, with less-robust service and response infrastructure."

INTRO/MAIN NARRATIVE

1. Line 8 to 15: The intro starts with SGBV against women and sets the tone as if this study were focused on women only. Please delete the the first two statements; it will not change much since there is rich epi data on SGBV provided down the line.

2. Line 26-27 rephrase: ..."with more than 56,000 confirmed cases as of ..."

3. Line 35-36: "as of November 2020, the number of people who can gather in groups **was** still limited." Please write this all in past tense as this is all in the past.

4. Page 7, line 24-29: "During the **ongoing** COVID-19 outbreak, seven (suggest several not seven, as time has passed) months of lockdown measures, economic challenges, health concerns, and changing global relations have increased concerns **of**" (again, this study is not limited to women only so allow the reader to have a wider thinking on this) a heightened risk of SGBV.

5. "We focus on sexual violence because it has received less attention during the pandemic compared to other forms of gender-based violence." I would temper this a bit, I believe the authors are referring to physical vs sexual violence, then name it and be specific with what you mean, because emotional/psychological violence is far less often discussed than sexual or physical.

6. "To our knowledge, no previous research has compared adults and children regarding the prevalence of violations committed by multiple perpetrators"-in Kenya, in African countries, or globally? Please specify.

METHODS

1. I highly commend the authors for not focusing the data collection only on women or children/girls, even though we eventually find out that most adult and child victims were female. It would have been of high additional value to do a simple analysis of number (if not some detail of victim sex/age and pattern) of reports received by these orgs in the study setting in say 2019 (pre COVID) vs 2020 (COVID)...and calculate percent changes. Can that info be provided?

2. "A quantitative between-group (added hyphen, deletes "s') prospective research design was used."

3. Participant section, line 15: Define the acronym WHO at first use. If only used once, no need to provide the acronym. Also add "years" to 18-72 in this section.

4. Ethics section: "The confidentiality of the data was maintained by the research team, and participants were not placed at risk of harm as a result of this study." I don't think you can make this statement as a researcher-that is a determination made by an IRB-or participants. Rather, briefly state how you attempted to maintain privacy and confidentiality, and in what way(s) was the Data Protection act adhered to?

5. Stats analysis-Line 17, please place comma after "attack". Line 19, add :statistical" before common-support problem-some readers may not immediately recognize this as stats terminology. Line 24-**could** not exploit this (use past tense). Line 33-supplemented (past tense)

6. "Values that fall between .40 and .60 are interpreted as large, whereas values that fall between .20 and .40 are interpreted as moderate in magnitude. Values

smaller than .20 are regarded as associations that are small in magnitude." Please put in past tense.

7. Line 45: Our data are freely available *at*:

RESULTS

1. Suggest deleting the first statement, rephrasing with more technical stats language : Bivariate (chi square) analysis indicates that... (Table 4). Children were also more likely to be attacked....Associations (add s) between age and private location...

2. Suggest more informative title for Table 4 eg Comparisons between characteristics of sexual violence against children vs adults: bivariate analysis. Also, use the more descriptive "Daytime Attack" vs Day, and Multiple Perpetrators (vs Multiple) for the labels. Please do same for Table 5

3. Try adding to or replacing chi-square results with "bivariate analysis"..eg bivariate analysis (chi square results)

4. Also please develop a more descriptive title for Table 5 other than Table of Coefficients

5. Any info on attacks (esp among adults) perpetuated by work colleagues even if in informal or formal work settings?

DISCUSSION

1. Comparisons to current literature-line 36 to 40. While it is stated in limitations, this statement should not be made without an immediate caution, that this was not a nationally representative sample, and violence was also happening in a particular crisis-not general-so it is not necessarily a statistically supported or valid observation.

2. "Further, in keeping with previous research, significantly more adults were violated by multiple perpetrators compared to children." Multiple perpetrators simultaneously in one attack or in multiple instances of attacks on the same survivor?

3. "This may reflect differential rates of victimization for men compared to boys." This should be further qualified to say that the age of the boys is a big factor here, as they more vulnerable due to youth and not necessarily sex, in the context of this statement.

4. Lines 17-26: excellent points raised re: issues with reporting sexual violence against males.

5. Line 28-30: "The children and adults in our sample were more likely to be violated by someone they knew than a stranger." I think authors meant to say the children only, as the very next statement is contradictory if "and adults" is not deleted.

6. Strengths and Limitations-line 29-31. Suggest: Further, our data do not provide information about **overall** sexual violence trends *in our study setting*. (If that is what you meant by this statement).

7. Line 40: "For example, the Demographic and Health Survey in Kenya, which is a nationally representative survey of adults occurring every five years, does not gather in-depth information

	about violations." Consider replacing "occurring" with "conducted". Also, specify exactly what you mean by the indepth info that the DHS has missed (which you may have also collected) 8. Reccs and Conclusions: First statement conveys the appropriate sentiment/recommendation, but is stated weakly. Need stronger rephrasing eg considering words like advocate, implement, integrate, prioritize, enact, enforce...and that legal recourse provided to survivors for more consistent perpetrator consequences/convictions. Something along those lines. 9. Reccs: We have a huge missed opportunity here, we need narrative on how your study can directly influence policy eg how would knowing the type of perpetrator, venue, time of day help to influence policy and prevent attacks, investigate/prosecute cases or better provide services and healing for survivors?
--	--

VERSION 1 – AUTHOR RESPONSE

Reviewer: 1
 Prof. Tessa Roseboom, Amsterdam UMC Location AMC, Amsterdam UMC Locatie AMC

Comment:
 The paper is well written and highlights the importance of more policies and protection measures to prevent any further damage to current and future generations due to the increase in violence.

I fully support the authors plea but have several suggestions to improve the impact of the paper.

First, the introduction highlights the gender inequality in violence exposure during crises. This is a relevant point but makes the reader think the paper will only address violence against women and children, while in actual fact the paper also addresses violence against men. Therefore I suggest to rewrite the introduction to make clear that both men and women can become victims of violence and that there is a gender equality.

Response:
 Thank you for your feedback. The introduction has been edited to include less material focusing on women’s experiences of violence. Additionally, we have noted that SGBV is an issue faced by both men and women, but that it does disproportionately impact women. Thus, much of the research evidence reflects this. The second line of the first page now reads:

Whilst SGBV affects women, men and children, it disproportionately affects women and girls, with one in three women having faced SGBV in their lifetimes worldwide.

Comment:
 Also, in rewriting the introduction, please make it more concise and shorter.

Response:
 Thank you for your feedback. The introduction has been condensed, and is now more focused and shorter.

Comment:

The authors report a clear increase in reports of violence but do not really address the extent to which this reflects both an actual increase in violence or also additionally a change in reporting. Are there any indications that there is less reporting so that the increase reported is only the tip of the iceberg of actual violence acts?

Response:

This is an important point and we are grateful to you for raising it. The proportion of cases of SGBV that are reported is in general extremely hard to measure due to the nature of the crimes committed and survivor's (potential) experience of the criminal justice system. It is important to note that it is not clear how rates of reporting have changed over time. One possibility is that the growth of the Survivors' Network in the last year means that reporting rates have increased, However, experience on the ground suggests that Covid-related restrictions are likely to have reduced reporting rates, meaning, that our results likely reflect only a the portion of violence that was reported. There is currently no accurate data on reporting rates, however; hence, one of our key recommendations is for the Kenyan government to enact the systematic collection of real time data to identify crime hotspots over time, and to enable the study of the nature of violence and survivors access to services.

Comment:

The reporting of violence is only one measure of violence occurring and some victims of violence actually die. So in addition to reporting the numbers of reports about violence it would be worthwhile to also mention any mortality due to violence in this report to get a more complete picture of violence in the country.

Response:

There were no cases of mortality in our sample, this has been noted in the manuscript. In particular lines 16-18 on page 16 now read as follows:

Although SGBV, such as domestic violence, has been linked to cases of domestic homicide in Kenya, there were no mortalities in our study sample (e.g., Mungai, 2019).

Comment:

Additionally, I would urge the authors to write more about what could be done to prevent a further escalation of violence. Not only paying attention to the victims but also trying to PREVENT further violence from happening. There should be mentioning of policy interventions that could contribute to preventing violence from occurring.

Response:

We are grateful for the reviewer for this excellent suggestion and we have rewritten the Recommendations and Conclusion section. This section now contains a number of explicit policy recommendations that focus on the need to a) make preventing SGBV a key part of future crisis responses, and b) the establishment of a network of safe-spaces. In particular we emphasise the need to prioritise financial resources to create these spaces and the importance of advocacy in ensuring funds are well-spent at the local level.

Comment:

In the statistical methods, the authors report about the predictive ability of their models. This seems rather inappropriate because the models were not built to predict violence, but merely to explain differences in

prevalence of violence according to age, gender and perpetrator, so please remove the comments about the predictive value.

Response:

Thank you for making this point. You are of course correct and comments about predictive value have been removed.

Comment:

This is an important paper that could gain power if the above suggestions are addressed.

Response:

Thank you. We believe that the paper has been much improved by incorporating your suggestions.

Reviewer: 2

Dr. Bharti Khurana, Harvard University

Response:

Thank you. In the re-written Recommendations and Conclusion section we now discuss the importance of local 'social halls' which can serve as a network of safe-spaces when schools are closed.

Reviewer: 3

Dr. N Sam-Agudu

Comment:

1. My major issue with this paper is the authorship credit. There is a problem at first glance, where for a study done exclusively in Kenya, one only sees one Kenyan/Kenya affiliated co-author, and in the 8th of 10 positions in the byline. Everyone else is from the UK. I would like the authors to reconsider the local contributing research staff included for authorship credit. The authors have described a cohesive consultative and implementation partnership between UK and Kenya-based staff. From the manuscript narratives, one can see that local Kenyan staff made significant contributions, at the very least in design and data collection." Please let more of that reflect in the author byline; more of the key contributing Kenyan staff may be invited to further contribute to the requested manuscript revisions, provide further critical review, and approve the final version to be submitted.

The authors themselves stress on how heavily they relied on the local advocacy and sexual violence survivor service/support groups in designing the tools and implementing the study. It is not clear to what extent these local groups or individual representatives) contributed to analyzing, interpreting and reporting the data, and whether they were invited to do so. However, the level of contribution of these local groups is quite obviously high. Authors also report that the data belong to the local WK foundation, if so, how come many more members of this data-providing foundation or key local Kenyans consulted did not contribute enough to warrant authorship, but several other people from the UK did?

Response:

Thank you for raising this and encouraging us to revisit the important issue of authorship. The issue of working in fair and equitable research partnerships is indeed an important matter that we have considered throughout the research process.

The secondary data we analyse was originally collected by a total of 87 human rights defenders who recorded the incidents as part of their work in assisting survivors access vital services, which was critical

and difficult work owing to the curfew. Their work is crucial, but also carries significant security risks. Curfews, lockdowns, and other measures create opportunities for people in authority to economically exploit and violate people. In Kenya, for example, post-election violence in 2007-2008 included many instances of rape or excessive force by police or protestors or individuals out past curfew. This pattern was repeated during the 2017 post-election period as well, where 24 people were killed in the days following the election, with at least six being killed in clashes with police. During the first ten days of Kenya's dusk-to-dawn curfew, six people died from police violence and there have been more reports of police extorting money from residents or taking bribes after arresting or kidnapping individuals, as well as rapes of persons in detention facilities. Human rights defenders operate in a precarious and dangerous environment.

Following our receipt of the reviews, the research team met with some of the human rights defenders to revisit the issue of authorship. We concluded from our discussion that it would not be safe to include the human rights defenders who collected the data as authors on the paper.

In redrafting the manuscript, we took the opportunity to involve more of our other local collaborators in the research and the paper is much improved for having done so. To reflect this Dorothy Njoroge (our collaborator on some of our ongoing SGBV projects in Kenya) and Catherine Kamau (who coordinated our receiving the data) joined the team of authors, and contributed to the writing of the policy sections of the manuscript. They are now both credited as authors. This also has meant we have been able to have a less unbalanced byline while also remaining consistent with the authorship principles outlined by the ICMJE. (<http://www.icmje.org/recommendations/browse/roles-and-responsibilities/defining-the-role-of-authors-and-contributors.html>)

Comment

2. There is no mention of the study period the reported data relates to; I have looked for it, and see none. At what period in the pandemic did the reports occur and the data collected? It is critically important, and should be clearly stated in the Abstract as well as the Methods section.

Response:

Thank you. The study period has been added to the abstract and methods sections. In particular, the Design statement now reads as follows:

Design A prospective cross-sectional research design was used with data collected from March-August 2020.

And the second sentence on page 9, in the Participants section, reads:

All were residents of Kenya, living in 23 counties, and had contacted human rights defenders for assistance in obtaining vital services in the aftermath of sexual violence during the COVID-19 pandemic between March and August 2020.

Comment:

TITLE: Suggest addingpatterns of **sexual** violence...The word sexual is critical

Response:

We agree and thank you for making this point. The title has been edited to reflect this and now is:

Patterns of Sexual Violence Against Adults and Children during the COVID-19 Pandemic in Kenya

Comments:

ABSTRACT

1. Design-add dates of study period
2. Results-add some actual hard data and stats, rather than simply describing. Sometimes the abstract is the only thing your audience will read.
3. "On average, the children in the sample were four years younger compared to the average age reported in national samples pre-pandemic (age 12 versus 16). Survivors were more likely to be female than male." Given that the study sample is not nationally representative, this statement has to be taken with a large grain of salt (and the authors have said so themselves)-as such, it really should not be herein the abstract. Please delete and use the space to provide harder evidence that was statistically significant and less vulnerable to the study's limitations.
4. Conclusions-... during the pandemic **and for future humanitarian crises**.

Response:

Thank you we have deleted the suggested text, and edited the abstract along the lines you suggest. It now reads as follows:

Results Bivariate analyses found that children were more likely than adults to be attacked during the daytime (59% vs. 44%, $p < .001$), by a single perpetrator rather than multiple perpetrators (13% vs. 31%, $p < .001$), in a private as opposed to a public location (66% vs. 45%, $p < .001$) and by someone known to the child (76% vs. 58%, $p < .001$). Children were violated most often by neighbours (29%) and family members (20%), whereas adults were equally likely to be attacked by strangers (41%) and persons known to them (59%). These variables were entered as predictors into a logistic regression model that significantly predicted the age group of the survivor, $\chi^2(5, N = 541) = 53.3, p = < .001$.

Conclusions Patterns of sexual violence against adult and child survivors during the COVID-19 pandemic are different, suggesting age-related measures are needed in national emergency plans to adequately address sexual violence during the pandemic and for future humanitarian crises.

Comment:

STRENGTHS AND LIMITATIONS

1. Lines 26-31-apt statement, agree, please correct "a representative" to just "representative" (delete "a"). Do the same for other instances where this shows up in the paper-about 2-3 more times

Response:

Thank you for spotting this typo. We have made the correction and the sentence now reads:

The sample was comprised of individuals who were seeking help in accessing vital services; therefore, inferences about patterns nationally in Kenya cannot be made because the data may not be representative.

Comment:

2. Line 36 to 43: This statement is true for everywhere in the world regardless of economics, may be more pressing in LMICs, so I suggest rephrasing: ..."about sexual violence **especially in countries like Kenya, with less-robust service and response infrastructure."

Thank you. We have amended the sentence as suggested. It now reads:

Information about whether patterns of sexual violence are changing during the pandemic remains unknown because sexual violence is underreported and also more difficult to report and document during emergencies/humanitarian situations/pandemics, and there is a need for real time data collection systems that gather and analyse detailed, longitudinal information about sexual violence especially in developing countries like Kenya, where service and response infrastructure are not as robust.

Comment:

INTRO/MAIN NARRATIVE

1. Line 8 to 15: The intro starts with SGBV against women and sets the tone as if this study were focused on women only. Please delete the first two statements; it will not change much since there is rich epi data on SGBV provided down the line.

Response:

As noted in our response to another reviewer above, we have changed this part of the intro to emphasise that SGBV also affects children and men. It now reads:

Whilst SGBV affects women, men and children, it predominantly affects women and girls, with one in three women having faced SGBV in their lifetimes worldwide.²

Comment:

2. Line 26-27 rephrase: ..."with more than 56,000 confirmed cases as of ..."

Response:

Thank you. This sentence now reads:

The arrival of COVID-19 in Kenya in early March 2020 marked the start of another national crisis, with more than 56,000 confirmed cases as of November 2020.

Comment:

3. Line 35-36: "as of November 2020, the number of people who can gather in groups **was** still limited." Please write this all in past tense as this is all in the past.

Response:

Thank you, we have amended as suggested. The relevant sentences now read:

These measures have been extended and modified multiple times, and by November 2020, the number of people who could gather in groups was still limited. Schools were also still closed, but universities opened, and air travel restrictions were lifted.⁶ While the measures have undoubtedly curbed the spread of the disease, they seemed to be compromising the safety and well-being of citizens.

Comment:

4. Page 7, line 24-29: "During the **ongoing** COVID-19 outbreak, seven (suggest several not seven, as time has passed) months of lockdown measures, economic challenges, health concerns, and changing global relations have increased concerns **of** (again, this study is not limited to women only so allow the reader to have an wider thinking on this) a heightened risk of SGBV.

Response:

We have amended as suggested. The relevant sentence now reads as follows:

During the ongoing COVID-19 outbreak, several months of lockdown measures, economic challenges, health concerns, and changing global relations have increased concerns of a heightened risk of SGBV.

Comment:

5. "We focus on sexual violence because it has received less attention during the pandemic compared to other forms of gender-based violence." I would temper this a bit, I believe the authors are referring to physical vs sexual violence, then name it and be specific with what you mean, because emotional/psychological violence is far less often discussed than sexual or physical.

Response:

Again, an excellent point. We have amended as suggested. The relevant sentence now reads:

We focus on sexual violence because it has received less attention during the pandemic compared to physical violence.

Comment:

6. "To our knowledge, no previous research has compared adults and children regarding the prevalence of violations committed by multiple perpetrators"-in Kenya, in African countries, or globally? Please specify.

Response:

This is indeed an important distinction. We have amended the relevant sentence to refer to Kenya. The manuscript has been edited to reflect that to our knowledge, there has been no research investigating the prevalence of multiple perpetrator incidences against adults or children in Kenya. The sentence now reads:

To our knowledge, no previous research has compared adults and children regarding the prevalence of violations committed by multiple perpetrators in Kenya. Hence, no age group predictions were made concerning multiple perpetrators.

Comment:

METHODS

1. I highly commend the authors for not focusing the data collection only on women or children/girls, even though we eventually find out that most adult and child victims were female.

It would have been of high additional value to do a simple analysis of number (if not some detail of victim sex/age and pattern) of reports received by these orgs in the study setting in say 2019 (pre COVID) vs 2020 (COVID)...and calculate percent changes. Can that info be provided?

Response:

We agree that a comparison would be extremely useful. Unfortunately, because of how data was collected during the pandemic (i.e. different reporting methods were used), there is no comparable data to reference from an earlier period.

Comment:

2. "A quantitative between-group (added hyphen, deletes "s") prospective research design was used."

Response:

Thank you. We have corrected this. The sentence now reads:

A quantitative between-group prospective research design was used. The criterion variable was age group (child or adult survivor).

Comment:

3. Participant section, line 15: Define the acronym WHO at first use. If only used once, no need to provide the acronym. Also add "years" to 18-72 in this section.

Response:

Both have been corrected. The relevant sentences now read:

The interview protocol was informed by World Health Organization's ethical principles for research on SGBV and safety protocols developed by the human rights defenders for conducting their work with survivors

And:

The participants in the final sample for the child group ranged in age from 8 months to 17 years ($M = 12.6$, $SD = 3.9$), 83% were girls, and 93% were perpetrated against by men, and for the adult group, ranged in age from 18 to 72 years ($M = 27.1$, $SD = 8.1$), 92% were women, and 96% were perpetrated against by men.

Comment:

4. Ethics section: "The confidentiality of the data was maintained by the research team, and participants were not placed at risk of harm as a result of this study." I don't think you can make this statement as a researcher-that is a determination made by an IRB-or participants. Rather, briefly state how you attempted to maintain privacy and confidentiality, and in what way(s) was the Data Protection act adhered to?

Response:

Thank you for your feedback. The manuscript has been edited to reflect how the DPA was adhered to and how the risk of harm to participants was mitigated. The relevant section now reads as follows:

The confidentiality of the data was maintained by the research team, and safety precautions were always take to minimize any risks that might cause physical harm to participants of this study. Data collection involved qualitative interviews only, and participants were offered psychological services post-interview. The Kenyan Data Protection Act (2019) was adhered to in the conduct of this research study.²⁰ Special attention was paid to Part IV of the Act, which notes that personal data should be processed with special attention to the privacy of the data subject, data should only be collected for specified and legitimate purposes, and that the data subject has a right to know how their data is used.

Comment:

5. Stats analysis-Line 17, please place comma after "attack". Line 19, add :statistical" before common-support problem-some readers may not immediately recognize this as stats terminology. Line 24-****could**** not exploit this (use past tense). Line 33-supplemented (past tense)

Response:

Thank you. We have made the suggested amendments. The relevant section now reads as follows:

While our data contain detailed information about each attack, we restrict our analysis to a limited number of binary variables in Table 1 as predictors. This is because there is a risk of a statistical common-support problem if we use a finer grained analysis. For example, whilst we have detailed data on where survivors were attacked, or their relationship with the perpetrator, we could not exploit this as few in the child group were attacked going to work, or by their spouse or partner. To avoid this difficulty, we used the coarser coding of relationship, known versus stranger perpetrator, and whether the attack took place in a public place or in private. This ensured that there were sufficient numbers of both children and adults in all categories. We then supplemented this analysis by tabulating the finer coding in Tables 2 and 3.

Comment:

6. "Values that fall between .40 and .60 are interpreted as large, whereas values that fall between .20 and .40 are interpreted as moderate in magnitude. Values smaller than .20 are regarded as associations that are small in magnitude." Please put in past tense.

Response:

Indeed. Thank you. This now reads:

Values that fall between .41 and .60 were interpreted as large, whereas values that fall between .20 and .40 were interpreted as moderate in magnitude. Values smaller than .20 were regarded as associations that are small in magnitude.

Comment:

7. Line 45: Our data are freely available *at*:

Response:

Thank you. We have made the correction, and this section now reads:

Our data are freely available at: <https://osf.io/b9dzp/>.

Comment:

RESULTS

1. Suggest deleting the first statement, rephrasing with more technical stats language : Bivariate (chi square) analysis indicates that... (Table 4). Children were also more likely to be attacked....Associations (add s) between age and private location...

Response:

Thank you for your feedback, the manuscript was edited to include more technical statistics language.

This section now reads:

Bivariate (chi square) analysis indicates that children compared to adults were less likely to be female and less likely to be attacked by multiple perpetrators (Table 4). Children were also more likely to be attacked in a private location, by a known perpetrator, and were more likely to be attacked in the daytime. The associations between age and private location, and age and multiple perpetrators was moderately large, whereas the strength of the other associations, while statistically significant, was small in magnitude.

Comment:

2. Suggest more informative title for Table 4 eg Comparisons between characteristics of sexual violence against children vs adults: bivariate analysis. Also, use the more descriptive "Daytime Attack" vs Day, and Multiple Perpetrators (vs Multiple) for the labels. Please do same for Table 5

Response:

Thank you for this suggestion. The table title and contents have been edited accordingly. They now read:

Comparisons between characteristics of sexual violence against children vs. adults: bivariate analysis

And:

Outputs of logistic regression by predictor variable

Comment:

3. Try adding to or replacing chi-square results with “bivariate analysis”..eg bivariate analysis (chi square results)

Response:

Thank you for this suggestion. This has been changed in the manuscript. The relevant sentence now reads:

In the logistic regression model, the variables that were statistically significant from the bivariate analysis (chi square results) were entered as predictors (i.e., female survivor, daytime attack, private versus public location, multiple perpetrators, and known versus stranger perpetrator), and the dependent variable was age group.

Comment:

4. Also please develop a more descriptive title for Table 5 other than Table of Coefficients

Response:

Thank you for the suggestion, the table title has been changed as stated above.

Comment:

5. Any info on attacks (esp among adults) perpetuated by work colleagues even if in informal or formal work settings?

Response:

This is an excellent suggestion. Unfortunately, while we collected data concerning whether the attacks occurred on the way to/from work, there is not enough data to separate attacks by work colleagues from other public attacks.

Comment:

DISCUSSION

1. Comparisons to current literature-line 36 to 40. While it is stated in limitations, this statement should not be made without an immediate caution, that this was not a nationally representative sample, and violence was also happening in a particular crisis-not general-so it is not necessarily a statistically supported or valid observation.

Response:

This is an important point. We have revised to make this clear:

Our sample was not nationally representative due to time and resource constraints, and it must also be noted that this violence was occurring within a particular crisis and the consequences of other crises may

be different. However, it is still notable that the child survivors in our sample were younger than previous national samples have indicated. A recent study in Kenya noted that survivors who are being seen in medical settings are younger pre-pandemic compared to prepandemic, speculating this was due to school closures. Although multiple studies have found that domestic violence in Kenya is often linked to cases of domestic homicide, there were no mortalities in our study sample.

Comment:

2. "Further, in keeping with previous research, significantly more adults were violated by multiple perpetrators compared to children." Multiple perpetrators simultaneously in one attack or in multiple instances of attacks on the same survivor?

Response

Thank you. We have edited the sentence to clarify that we mean multiple perpetrators in one incident. The sentence now reads:

Further, in keeping with previous research, significantly more adults were violated by multiple perpetrators in one attack compared to children.

Comment:

3. "This may reflect differential rates of victimization for men compared to boys." This should be further qualified to say that the age of the boys is a big factor here, as they more vulnerable due to youth and not necessarily sex, in the context of this statement.

Response:

Thank you. We have revised as suggested. It now reads:

This may reflect differential rates of victimization for men compared to boys, as boys are more vulnerable to assault than men due to their age.

Comment:

4. Lines 17-26: excellent points raised re: issues with reporting sexual violence against males.

Thank you.

Comment:

5. Line 28-30: "The children and adults in our sample were more likely to be violated by someone they knew than a stranger." I think authors meant to say the children only, as the very next statement is contradictory if "and adults" is not deleted.

Response:

Indeed. Thank you for highlighting this. The sentence has been changed to reflect only children, it now reads:

The children in our sample were more likely to be violated by someone they knew than by a stranger.

Comment:

6. Strengths and Limitations-line 29-31. Suggest: Further, our data do not provide information about ****overall**** sexual violence trends ***in our study setting***. (If that is what you meant by this statement).

Response:

Thank you. We have made the correction and it now reads:
Further, our data do not provide information about overall sexual violence trends in our study setting.

Comment:

7. Line 40: "For example, the Demographic and Health Survey in Kenya, which is a nationally representative survey of adults occurring every five years, does not gather in-depth information about violations." Consider replacing "occurring" with "conducted". Also, specify exactly what you mean by the indepth info that the DHS has missed (which you may have also collected)

Response:

Thank you for your feedback, more information has been added about the missing data from the DHS. The relevant text now reads:

For example, the Demographic and Health Survey in Kenya, which is a nationally representative survey of adults conducted every five years, does not gather in-depth information about violations. For example, it does not collect information about the time of day (or night) the attack occurred, if there were multiple perpetrators involved in the attack, and if the attack took place in a public or private location.

Comment:

8. Reccs and Conclusions: First statement conveys the appropriate sentiment/recommendation, but is stated weakly. Need stronger rephrasing eg considering words like advocate, implement, integrate, prioritize, enact, enforce...and that legal recourse provided to survivors for more consistent perpetrator consequences/convictions. Something along those lines.

9. Reccs: We have a huge missed opportunity here, we need narrative on how your study can directly influence policy eg how would knowing the type of perpetrator, venue, time of day help to influence policy and prevent attacks, investigate/prosecute cases or better provide services and healing for survivors?

Response:

Thank you for both of these comments. We have substantially revised the final section of the paper along the lines you suggest. We now discuss a) recommendations for how the prevention of SGBV should be introduced into national crisis policy;b) recommendations for real-time data collection; c) the ultimate necessity of changing societal perceptions of SGBV; and d) How the particular needs of survivors must be provided for during crises. In so doing we have sought to use unequivocal language as suggested. A sentence has been added about legal recourse and increased convictions. For your convenience we reproduce the entire section below. It now reads:

We urge policy makers to ensure that government COVID-19 emergency management and recovery planning adequately addresses SGBV and that minimising the risk of additional SGBV risk is integrated into national crisis policies. In particular, the results suggest this should include the provision of adequate alternative safe spaces and shelters when schools are closed. Further, many communities have voluntarily-organized neighbourhood watch groups that are focused on security issues, and these should be explicitly expanded and supported to monitor and prevent SGBV. Community leaders have also said that there is a need for more social halls – community facilities for holding meetings, screening educational films, and other social activities. These structures can be a safe space for children and can be built using constituency development funds, which each Member of Parliament in Kenya receives to undertake projects that will address the urgent needs of their constituents.

Our results indicate the importance of high-quality and timely data in understanding and thus combating SGBV. We recommend governments invest in real-time data collection and analysis systems to capture the evolving distribution of SGBV and to allow for the study of regional trends. Data would allow authorities to identify crime hot spots and violations being perpetrated by serial offenders, and to monitor the accessibility of vital services to help ensure that survivors have support. This information is crucial in designing effective interventions. For example, by knowing the location and time of attacks, responsive programmes could be put in place to engage children in other activities (e.g., drama, sports, and other educational and recreation activities) when they are not in school during the day to ensure that they are not left unattended by a responsible adult. These interventions can be low cost, with communities mobilized to create such activities with the help of university students, local NGOs, neighbourhood teachers, and religious organizations. Police patrols and community initiatives could also be planned for times at which SGBV rates peak to deter attacks and apprehend offenders. Further, the installation of street lighting might deter perpetrators from attacking women and children. Another suggestion is to establish a national sexual offender register in Kenya that would warn communities about high-risk offenders. The collection of real-time data can also inform educational programs that sensitize parents and children about community risks. These efforts must be survivor-centred, involving survivors in the implementation and evaluation of the systems.

More generally, the results in this paper highlight the latent risk of SGBV, particularly for women and girls. While its manifestation currently waxes and wanes dependent on the context, meaningful reductions in violence will require changing the narrative such that SGBV is understood to be a crime, a gross violation of human rights, and its pre-eminent importance as a determinant of physical, emotional, and mental health is reflected in national and county budgets. Funding for programming, interventions, and research should be included.

High rates of SGBV also necessitate adequate protection for the needs of survivors. To this end, the national government has approved the use of the National Government Affirmative Fund to facilitate the establishment of safe spaces or shelters in all 47 counties to ensure survivors' safety and security is safe guarded. However, advocacy is required to ensure the funds are directed appropriately.

The implementation of emergency referral pathways that enable survivors to access comprehensive care and support services should be enacted by the government. Curfews and other social distancing regulations need to include SGBV response mechanisms to ensure the continued availability and accessibility of services for survivors. Further, the medico-legal response to SGBV can be strengthened by expediting restraining orders and prosecutions, and by establishing 'one-stop' centres to allow survivors to access essential services, and authorities to collect evidence, all in one location. This would also facilitate the preservation of evidence and protection of survivors to facilitate access to justice.

VERSION 2 – REVIEW

REVIEWER	Sam-Agudu, N Institute of Human Virology
REVIEW RETURNED	06-Aug-2021
GENERAL COMMENTS	Many thanks to the authors for their revisions. The addition and recognition of two more indigenous co-authors is noted and appreciated. It better reflects the cooperative work done by local and international partners to generate the data and manuscript.

	The content revisions are also noted and make the paper and its message much stronger. A few notes: GENERAL 1. Strengths and limitations section and elsewhere: please replace the term "developing country" with "resource-limited country/setting." ABSTRACT: 1. "...by a single perpetrator rather than multiple perpetrators (13% vs. 31%, $p < .001$)", should be 31% vs 13%, order should be flipped. RESULTS: 1. Suggest: "Bivariate (chi square) analysis indicates that compared to adults, child victims/survivors were less likely to be female and less likely to be attacked by multiple perpetrators (Table 4)." I would add something along the lines of the following statement to reinforce the findings that all victims/survivors were still overwhelmingly female. "Regardless, both child and adult victims were overwhelmingly female; 83% and 92%, respectively." DISCUSSION 1. "Although SGBV, such as domestic violence, has been linked to cases of domestic homicide in Kenya, there were no mortalities *captured* in our study sample." 2. "For example, by knowing the location and time of attacks, responsive programmes could be put in place to engage children in other activities (e.g., drama, sports, and other educational and recreation activities) when they are not in school during the day to ensure that they are not left unattended by a responsible adult." I would ask the authors to reconsider this statement, as it places the burden heavily on parents and children to prevent/avoid attacks, rather than deterring perpetrators-many of whom are familiar people. It can be restated to portray higher vigilance and awareness of SGBV against children, believing children and investigating (rather than silence) when they report these events, and also getting educated on the definitions and consequences of SGBV and watching out for signs of SGBV among their children. REFS A few of the refs have incomplete information or the formatting for the author names (especially agencies as authors) are truncated likely due to formatting errors in the Reference Manager. Please correct these errors and complete the refs with DOIs and URLs for e-pubs and for webpage refs, respectively. For example, Refs 2, 3, 10, 20, 22, 23.
--	--

VERSION 2 – AUTHOR RESPONSE

Reviewer: 3

Dr. N Sam-Agudu, Institute of Human Virology, Institute of Human Virology Nigeria

Response:

Thank you for this feedback, the term 'developing country' has been removed in both locations and replaced with 'resource-limited country' to read:

longitudinal information about sexual violence especially in resource-limited countries like Kenya, where service and response infrastructure are not as robust.

And

which leads to underreporting, especially in resource-limited countries that have high levels of gender inequality

Comment:

"...by a single perpetrator rather than multiple perpetrators (13% vs. 31%, $p < .001$)", should be 31% vs 13%, order should be flipped.

Response:

Thank you for making this point, the percentages have been flipped.

Comment:

Suggest: "Bivariate (chi square) analysis indicates that compared to adults, child victims/survivors were less likely to be female and less likely to be attacked by multiple perpetrators (Table 4)." I would add something along the lines of the following statement to reinforce the findings that all victims/survivors were still overwhelmingly female. "Regardless, both child and adult victims were overwhelmingly female; 83% and 92%, respectively."

Response

Thank you for making this point. The text now reads:

Bivariate (chi square) analysis indicates that children compared to adults were less likely to be female and less likely to be attacked by multiple perpetrators (Table 4). Regardless, both child and adult victims were overwhelmingly female; 83% and 92%, respectively.

Comment:

*"Although SGBV, such as domestic violence, has been linked to cases of domestic homicide in Kenya, there were no mortalities *captured* in our study sample."*

Response:

Thank you for making this point, we have amended the sentence which now reads:

Although SGBV, such as domestic violence, has been linked to cases of domestic homicide in Kenya, there were no mortalities captured in our study sample

Comment

"For example, by knowing the location and time of attacks, responsive programmes could be put in place to engage children in other activities (e.g., drama, sports, and other educational and recreation activities) when they are not in school during the day to ensure that they are not left unattended by a responsible adult." I would ask the authors to reconsider this statement, as it places the

burden heavily on parents and children to prevent/avoid attacks, rather than deterring perpetrators-many of whom are familiar people. It can be restated to portray higher vigilance and awareness of SGBV against children, believing children and investigating (rather than silence) when they report these events, and also getting educated on the definitions and consequences of SGBV and watching out for signs of SGBV among their children.

Response:

Thank you for bringing up this point. We have edited the text to read:

For example, by knowing the location and time of attacks, there can be more vigilance and awareness of SGBV against children. Additionally, this information can be used to provide further education about SGBV against children and can highlight signs to look out for of abuse.

Comment

A few of the refs have incomplete information or the formatting for the author names (especially agencies as authors) are truncated likely due to formatting errors in the Reference Manager. Please correct these errors and complete the refs with DOIs and URLs for e-pubs and for webpage refs, respectively. For example, Refs 2, 3, 10, 20, 22, 23.

Response:

Thank you for spotting this error. References have been edited to include full organization name for authors and to include URLs and DOIs as needed.